# Additive function approximation in the brain

**Kameron Decker Harris**
Paul G. Allen School of Computer Science and Engineering, Department of Biology
University of Washington
kamdh@uw.edu

## Abstract

Many biological learning systems such as the mushroom body, hippocampus, and cerebellum are built from sparsely connected networks of neurons. For a new understanding of such networks, we study the function spaces induced by sparse random features and characterize what functions may and may not be learned. A network with $d$ inputs per neuron is found to be equivalent to an additive model of order $d$, whereas with a degree distribution the network combines additive terms of different orders. We identify three specific advantages of sparsity: additive function approximation is a powerful inductive bias that limits the curse of dimensionality, sparse networks are stable to outlier noise in the inputs, and sparse random features are scalable. Thus, even simple brain architectures can be powerful function approximators. Finally, we hope that this work helps popularize kernel theories of networks among computational neuroscientists.

## 1   Introduction

Kernel function spaces are popular among machine learning researchers as a potentially tractable framework for understanding artificial neural networks trained via gradient descent [e.g. 1, 2, 3, 4, 5, 6]. Artificial neural networks are an area of intense interest due to their often surprising empirical performance on a number of challenging problems and our still incomplete theoretical understanding. Yet computational neuroscientists have not widely applied these new theoretical tools to describe the ability of biological networks to perform function approximation.

The idea of using fixed random weights in a neural network is primordial, and was a part of Rosenblatt's perceptron model of the retina [7]. Random features have then resurfaced under many guises: random centers in radial basis function networks [8], functional link networks [9], Gaussian processes (GPs) [10, 11], and so-called extreme learning machines [12]; see [13] for a review. Random feature networks, where the neurons are initialized with random weights and only the readout layer is trained, were proposed by Rahimi and Recht in order to improve the performance of kernel methods [14, 15] and can perform well for many problems [13].

In parallel to these developments in machine learning, computational neuroscientists have also studied the properties of random networks with a goal towards understanding neurons in real brains. To a first approximation, many neuronal circuits seem to be randomly organized [16, 17, 18, 19, 20]. However, the recent theory of random features appears to be mostly unknown to the greater computational neuroscience community.

Here, we study random feature networks with *sparse connectivity*: the hidden neurons each receive input from a random, sparse subset of input neurons. This is inspired by the observation that the connectivity in a variety of predominantly feedforward brain networks is approximately random and sparse. These brain areas include the cerebellar cortex, invertebrate mushroom body, and dentate gyrus of the hippocampus [21]. All of these areas perform pattern separation and associative learning. The cerebellum is important for motor control, while the mushroom body and dentate gyrus are

33rd Conference on Neural Information Processing Systems (NeurIPS 2019), Vancouver, Canada.

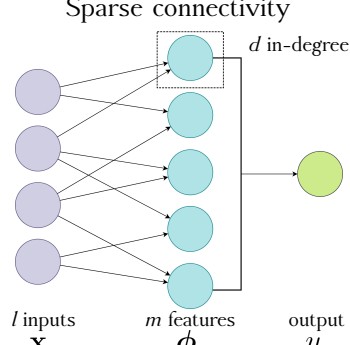

- Additivity as inductive bias
- Stability to input noise
- Scalable wiring & computation

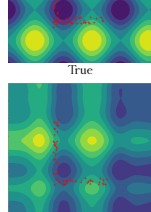

Example right: Learning the additive function

$$f(x_1, x_2) = \sin(x_1) + \sin(x_2)$$

from limited data using sparse random features

Figure 1: Sparse connectivity in a shallow neural network. The function shown is the sparse random feature approximation to an additive sum of sines, learned from poorly distributed samples (red crosses). Additivity offers structure which may be leveraged for fast and efficient learning.

general learning and memory areas for invertebrates and vertebrates, respectively, and may have evolved from a similar structure in the ancient bilaterian ancestor [22]. Recent work has argued that the sparsity observed in these areas may be optimized to balance the dimensionality of representation with wiring cost [20]. Sparse connectivity has been used to compress artificial networks and speed up computation [23, 24, 25], whereas convolutions are a kind of structured sparsity [26, 27].

We show that sparse random features approximate additive kernels [28, 29, 30, 31] with arbitrary orders of interaction. The in-degree of the hidden neurons $d$ sets the order of interaction. When the degrees of the neurons are drawn from a distribution, the resulting kernel contains a weighted mixture of interactions. These sparse features offer advantages of generalization in high-dimensions, stability under perturbations of their input, and computational and biological efficiency.

## 2   Background: Random features and kernels

Now we will introduce the mathematical setting and review how random features give rise to kernels. The simplest artificial neural network contains a single hidden layer, of size $m$, receiving input from a layer of size $l$ (Figure 1). The activity in the hidden layer is given by, for $i \in [m]$,

$$\phi_i(\mathbf{x}) = h(\mathbf{w}_i^\mathsf{T}\mathbf{x} + b_i). \tag{1}$$

Here each $\phi_i$ is a feature in the hidden layer, $h$ is the nonlinearity, $\mathbf{W} = (\mathbf{w}_1, \mathbf{w}_2, \ldots, \mathbf{w}_m) \in \mathbb{R}^{l \times m}$ are the input to mixed weights, and $\mathbf{b} \in \mathbb{R}^m$ are their biases. We can write this in vector notation as $\boldsymbol{\phi}(\mathbf{x}) = h(\mathbf{W}^\mathsf{T}\mathbf{x} - \mathbf{b})$, where $\boldsymbol{\phi} : \mathbb{R}^l \to \mathbb{R}^m$.

Random features networks draw their input-hidden layer weights at random. Let the weights $\mathbf{w}_i$ and biases $b_i$ in the feature expansion (1) be sampled i.i.d. from a distribution $\mu$ on $\mathbb{R}^{l+1}$. Under mild assumptions, the inner product of the feature vectors for two inputs converges to its expectation

$$\frac{1}{m}\boldsymbol{\phi}(\mathbf{x})^\mathsf{T}\boldsymbol{\phi}(\mathbf{x}') \xrightarrow{m \to \infty} \mathbb{E}\left[\phi(\mathbf{x})\phi(\mathbf{x}')\right] = \int h(\mathbf{w}^\mathsf{T}\mathbf{x} + b)h(\mathbf{w}^\mathsf{T}\mathbf{x}' + b)\,\mathrm{d}\mu(\mathbf{w}, b) := k(\mathbf{x}, \mathbf{x}'). \tag{2}$$

We identify the limit (2) with a reproducing kernel $k(\mathbf{x}, \mathbf{x}')$ induced by the random features, since the limiting function is an inner product and thus always positive semidefinite [14]. The kernel defines an associated reproducing kernel Hilbert space (RKHS) of functions. For a finite network of width $m$, the inner product $\frac{1}{m}\boldsymbol{\phi}(\mathbf{x})^\mathsf{T}\boldsymbol{\phi}(\mathbf{x}')$ is a randomized approximation to the kernel $k(\mathbf{x}, \mathbf{x}')$.

## 3   Sparsely connected random feature kernels

We now turn to our main result: the general form of the random feature kernels with sparse, independent weights. For simplicity, we start with a regular model and then generalize the result to networks with varying in-degree. Two kernels that can be computed in closed form are highlighted.

Fix an in-degree $d$, where $1 \leq d \leq l$, and let $\mu | d$ be a distribution on $\mathbb{R}^d$ which induce, together with some nonlinearity $h$, the kernel $k_d(\mathbf{z}, \mathbf{z}')$ on $\mathbf{z}, \mathbf{z}' \in \mathbb{R}^d$ (for the moment, $d$ is not random). Sample a

sparse feature $i \in [m]$ in two steps: First, pick $d$ neighbors from all $\binom{l}{d}$ uniformly at random. Let $\mathcal{N}_i \subseteq [l]$ denote this set of neighbors. Second, sample $w_{ji} \sim \mu|d$ if $j \in \mathcal{N}_i$ and otherwise set $w_{ji} = 0$. We find that the resulting kernel

$$k_d^{\mathrm{reg}}(\mathbf{x}, \mathbf{x}') = \mathbb{E}[\mathbb{E}[\phi(\mathbf{x}_\mathcal{N})\phi(\mathbf{x}'_\mathcal{N})|\mathcal{N}]] = \binom{l}{d}^{-1} \sum_{\mathcal{N}:|\mathcal{N}|=d} k_d(\mathbf{x}_\mathcal{N}, \mathbf{x}'_\mathcal{N}). \tag{3}$$

Here $\mathbf{x}_\mathcal{N}$ denotes the length $d$ vector of $\mathbf{x}$ restricted to the neighborhood $\mathcal{N}$, with the other $l - d$ entries in $\mathbf{x}$ ignored.

More generally, the in-degrees may be chosen independently according to a *degree distribution*, so that $d$ becomes a random variable. Let $D(d)$ be the probability mass function of the hidden node in-degrees. Conditional on node $i$ having degree $d_i$, the in-neighborhood $\mathcal{N}_i$ is chosen uniformly at random among the $\binom{l}{d_i}$ possible sets. Then the induced kernel becomes

$$k_D^{\mathrm{dist}}(\mathbf{x}, \mathbf{x}') = \mathbb{E}[\mathbb{E}[\phi(\mathbf{x}_\mathcal{N})\phi(\mathbf{x}'_\mathcal{N})|\mathcal{N}, d]] = \sum_{d=0}^{l} D(d)\, k_d^{\mathrm{reg}}(\mathbf{x}, \mathbf{x}'). \tag{4}$$

For example, if every layer-two node chooses its inputs independently with probability $p$, the $D(d_i)$ is the probability mass function of the binomial distribution $\mathrm{Bin}(l, p)$. The regular model (3) is a special case of (4) with $D(d') = \mathbb{I}\{d' = d\}$. Extending the proof techniques in [14] yields:

**Claim** *The random map* $\frac{1}{m}\boldsymbol{\phi}(\mathbf{x})^\mathsf{T}\boldsymbol{\phi}(\mathbf{x}')$ *with $\kappa$-Lipschitz nonlinearity uniformly approximates* $k_D^{\mathrm{dist}}(\mathbf{x}, \mathbf{x}')$ *to error $\epsilon$ using* $m = O(\frac{l\kappa^2}{\epsilon^2}\log\frac{C}{\epsilon})$ *many features (the proof is contained in Appendix C).*

**Two simple examples** With Gaussian weights and regular $d = 1$, we find that (see Appendix B)

$$k_1^{\mathrm{reg}}(\mathbf{x}, \mathbf{x}') = 1 - \frac{1}{l}\|\mathrm{sgn}(\mathbf{x}) - \mathrm{sgn}(\mathbf{x}')\|_0 \qquad \text{if } h = \text{step function, and} \tag{5}$$

$$k_1^{\mathrm{reg}}(\mathbf{x}, \mathbf{x}') = 1 - \frac{c}{l}\|\mathbf{x} - \mathbf{x}'\|_1 \qquad \text{if } h = \text{sign function.} \tag{6}$$

## 4 Advantages of sparse connectivity

### 4.1 Additive modeling

The regular degree kernel (3) is a sum of kernels that only depend on combinations of $d$ inputs, making it an *additive kernel* of order $d$. The general expression for the degree distribution kernel (4) illustrates that sparsity leads to a mixture of additive kernels of different orders. These have been referred to as additive GPs [30], but these kind of models have a long history as generalized additive models [e.g. 28, 32]. For the regular degree model with $d = 1$, the sum in (3) is over neighborhoods of size one, simply the individual indices of the input space. Thus, for any two input neighborhoods $\mathcal{N}_1$ and $\mathcal{N}_2$, we have $|\mathcal{N}_1 \cap \mathcal{N}_2| = \emptyset$, and the RKHS corresponding to $k_1^{\mathrm{reg}}(\mathbf{x}, \mathbf{x}')$ is the direct sum of the subspaces $\mathcal{H} = \mathcal{H}_1 \oplus \ldots \oplus \mathcal{H}_l$. Thus regular $d = 1$ defines a first-order additive model, where $f(\mathbf{x}) = f_1(x_1) + \ldots + f_l(x_l)$. When $d > 1$ we allow interactions between subsets of $d$ variables, e.g. regular $d = 2$ leads to $f(\mathbf{x}) = f_{12}(x_1, x_2) + \ldots + f_{l-1,l}(x_{l-1}, x_l)$, all pairwise terms. These interactions are defined by the structure of the terms $k_d(\mathbf{x}_\mathcal{N}, \mathbf{x}'_\mathcal{N})$. Finally, the degree distribution $D(d)$ determines how much weight to place on different degrees of interaction.

**Generalization from fewer examples in high dimensions** Stone proved that first-order additive models do not suffer from the curse of dimensionality [33, 34], as the excess risk does not depend on the dimension $l$. Kandasamy and Yu [31] extended this result to $d$th-order additive models and found a bound on the excess risk of $O(l^{2d}n^{\frac{-2s}{2s+d}})$ or $O(l^{2d}C^d/n)$ for kernels with polynomial or exponential eigenvalue decay rates ($n$ is the number of samples and the constants $s$ and $C$ parametrize rates). Without additivity, these weaken to $O(n^{\frac{-2s}{2s+l}})$ and $O(C^l/n)$, much worse when $l \gg d$.

**Similarity to dropout** Dropout regularization [35, 36] in deep networks has been analyzed in a kernel/GP framework [37], leading to (4) with $D = \mathrm{Bin}(l, p)$ for a particular base kernel. Dropout may thus improve generalization by enforcing approximate additivity, for the reasons above.

## 4.2 Stability: robustness to noise or attacks affecting a few inputs

Equations (5) and (6) are similar: They differ only by the presence of an $\ell^0$-"norm" versus an $\ell^1$-norm and the presence of the sign function. Both norms are stable to outlying coordinates in an input $\mathbf{x}$. This property also holds for different nonlinearities and $1 < d \ll l$, since every feature $\phi_i(\mathbf{x})$ only depends on $d$ inputs, and therefore only a minority of the $m$ features will be affected by the few outliers.[1] Sufficiently sparse features will then be less affected by sparse noise than a fully-connected network, offering denoising advantages [e.g. 20]. A regressor $f(\mathbf{x}) = \boldsymbol{\alpha}^\mathsf{T}\boldsymbol{\phi}(\mathbf{x})$ built from these features is stable so long as $\|\boldsymbol{\alpha}\|_p$ is small, since $|f(\mathbf{x}) - f(\mathbf{x}')| \leq \|\boldsymbol{\alpha}\|_p\|\boldsymbol{\phi}(\mathbf{x}) - \boldsymbol{\phi}(\mathbf{x}')\|_q$ for any Hölder conjugates $1/p + 1/q = 1$. Thus if $\mathbf{x}' = \mathbf{x} + \mathbf{e}$ where $\mathbf{e}$ contains a small number of nonzero entries, then $f(\mathbf{x}') \approx f(\mathbf{x})$ since $\boldsymbol{\phi}(\mathbf{x}) \approx \boldsymbol{\phi}(\mathbf{x}')$. Stability also may guarantee the robustness of these networks to sparse adversarial attacks [38, 39, 40], although exactly the conditions under which these approximations hold ($p = \infty, q = 1$ is an interesting case) we leave for future work.

## 4.3 Scalability: computational and biological

**Computational**  Sparse random features give potentially huge improvements in scaling. Direct implementations of additive models incur a large cost for $d > 1$, since (3) requires a sum over $\binom{l}{d} = O(l^d)$ neighborhoods.[2] This leads to $O(n^2 l^d)$ time to compute the Gram matrix of $n$ examples and $O(n l^d)$ operations to evaluate $f(\mathbf{x})$. In our case, since the random features method is primal, we need to perform $O(nmd)$ computations to evaluate the feature matrix and the cost of evaluating $f(\mathbf{x})$ remains $O(md)$.[3] Sparse matrix-vector multiplication makes evaluation faster than the $O(ml)$ time it takes when connectivity is dense. For ridge regression, we have the usual advantages that computing an estimator takes $O(nm^2 + nmd)$ time and $O(nm + md)$ memory, rather than $O(n^3)$ time and $O(n^2)$ memory for a naïve kernel ridge method.

**Biological**  In a small animal such as a flying insect, space is extremely limited. Sparsity offers a huge advantage in terms of wiring cost [20]. Additive approximation also means that such animals can learn much more quickly, as seen in the mushroom body [41, 42, 43]. While the previous computational points do not apply as well to biology, since real neurons operate in parallel, fewer operations translate into lower metabolic cost for the animal.

## 5  Discussion

Inspired by their ubiquity in biology, we have studied sparse random networks of neurons using the theory of random features, finding the advantages of additivity, stability, and scalability. This theory shows that sparse networks such as those found in the mushroom body, cerebellum, and hippocampus can be powerful function approximators. Kernel theories of neural circuits may be more broadly applicable in the field of computational neuroscience.

**Expanding the theory of dimensionality in neuroscience**  Learning is easier in additive function spaces because they are *low-dimensional*, a possible explanation for few-shot learning in biological systems. Our theory is complementary to existing theories of dimensionality in neural systems [16, 44, 45, 46, 47, 20, 48, 49, 50], which defined dimensionality using a skewness measure of covariance eigenvalues. Kernel theory extends this concept, measuring dimensionality similarly [51] in the space of nonlinear functions spanned by the kernel.

**Limitations**  We model biological neurons as simple scalar functions, completely ignoring time and neuromodulatory context. It seems possible that a kernel theory could be developed for time- and context-dependent features. Our networks suppose i.i.d. weights, but weights that follow Dale's law should also be considered. We have not studied the sparsity of activity, postulated to be relevant in cerebellum. It remains to be demonstrated how the theory can make concrete, testable predictions, e.g. whether this theory may explain identity versus concentration encoding of odors or the discrimination/generalization tradeoff under experimental conditions.

---

[1] If one coordinate of $\mathbf{x}$ is noisy, the probability that the $i$th neuron is affected is $d_i/l \ll 1$.
[2] There is a more efficient method when working with a *tensor product kernel*, as in [29, 30, 31].
[3] Note that we need to take $m = \Omega(l)$ to ensure good approximation of the kernel (Appendix C).

**Acknowledgments** KDH was supported by a Washington Research Foundation postdoctoral fellowship. Thank you to Rajesh Rao for support during this project and to Bing Brunton for support and many helpful comments.

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

# Appendices: Additive function approximation in the brain

**Table of contents**

## A  Test problems

We have implemented sparse random features in Python to demonstrate the properties of learning in this basis. Our code, which provides a `scikit-learn` style `SparseRFRegressor` and `SparseRFClassifier` estimators, is available from https://github.com/kharris/sparse-random-features.

### A.1  Additive function approximation

#### A.1.1  Comparison with datasets from Kandasamy and Yu [1]

As said in the main text, Kandasamy and Yu [1] created a theory of the generalization properties of higher-order additive models. They supplemented this with an empirical study of a number of datasets using their Shrunk Additive Least Squares Approximation (SALSA) implementation of the additive kernel ridge regression (KRR). Their data and code were obtained from https://github.com/kirthevasank/salsa.

We compared the performance of SALSA to the sparse random feature approximation of the same kernel. We employ random sparse Fourier features with Gaussian weights $N(0, \sigma^2 \mathbf{I})$ with $\sigma = 0.05 \cdot \sqrt{d} n^{1/5}$ in order to match the Gaussian radial basis function used by Kandasamy and Yu [1]. We use $m = 300l$ features for every problem, with regular degree $d$ selected equal to the one chosen by SALSA. The regressor on the features is cross-validated ridge regression (`RidgeCV` from `scikit-learn`) with ridge penalty selected from 5 logarithmically spaced points between $10^{-4} \cdot n$ and $10^2 \cdot n$.

In Figure 2, we compare the performance of sparse random features to SALSA. Generally, the training and testing errors of the sparse model are slightly higher than for the kernel method, except for the `forestfires` dataset.

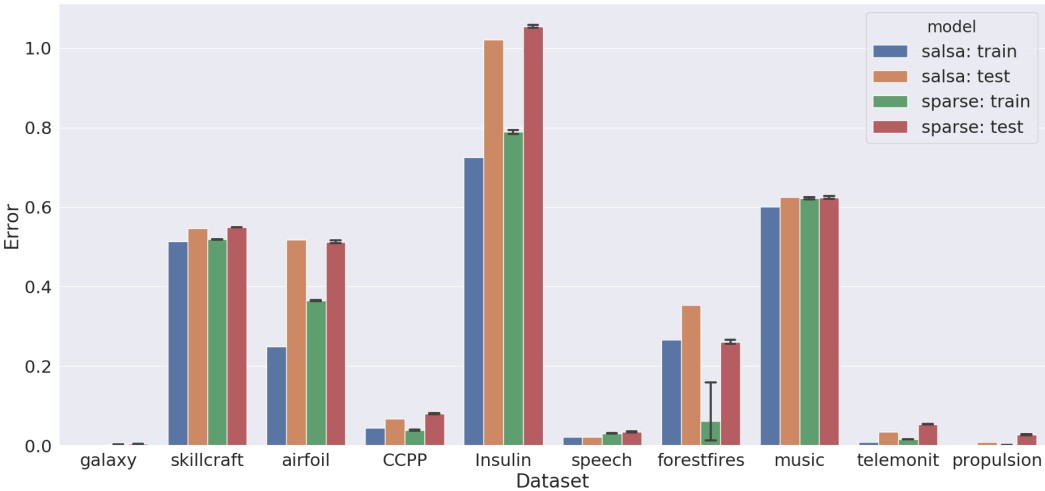

Figure 2: Comparison of sparse random feature approximation to additive kernel method SALSA [1]. The parameters were matched between the two models (see text). The sparse feature approximation performs slightly worse than the exact method, but similar.

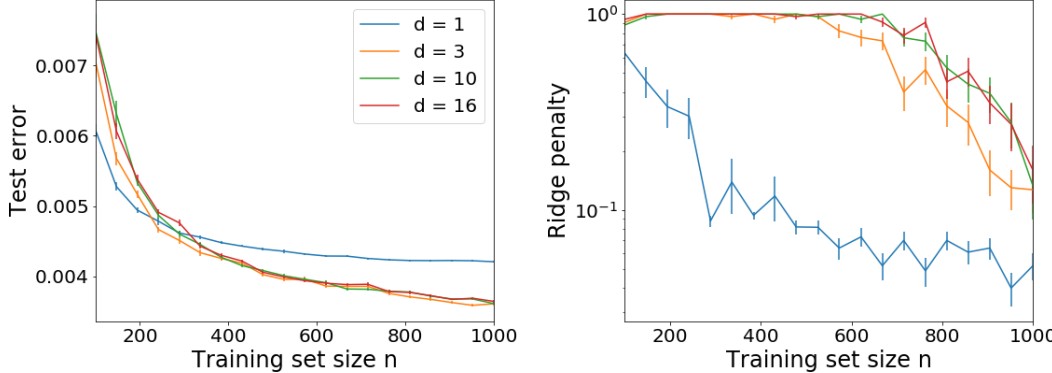

Figure 3: Performance of sparse random features of differing degree $d$ and training size $n$ for the polynomial test function. Test error is measured as mean square error with noise floor at 0.0025. As the amount of training data increases, higher $d$, i.e. more model complexity, is preferred.

### A.1.2 Polynomial test function shows generalization from fewer examples

We studied the speed of learning for a test function as well. The function to be learned $f(\mathbf{x})$ was a sparse polynomial plus a linear term:

$$f(\mathbf{x}) = c_1 \mathbf{a}^\mathsf{T} \mathbf{x} + c_2 \, p(\mathbf{x}).$$

The linear term took $\mathbf{a} \sim N(0, \mathbf{I})$, the polynomial $p$ was chosen to have 3 terms of degree 3 with weights drawn from $N(0, 1)$. The inputs $\mathbf{x}$ are drawn from the uniform distribution over $[0, 1]^{16}$. Gaussian noise $\epsilon$ with variance $0.05^2$ was added to generate observations $y_i = f(\mathbf{x}_i) + \epsilon_i$. Constants $c_1$ and $c_2$ were tuned by setting $c_1 = \frac{1}{\sigma_{\text{lin}}} \frac{1-\alpha}{\sqrt{\alpha^2 + (1-\alpha)^2}}$ and $c_2 = \frac{1}{\sigma_{\text{nonlin}}} \frac{\alpha}{\sqrt{\alpha^2 + (1-\alpha)^2}}$, where $\alpha = 0.05$ and $\sigma_{\text{lin}}$ and $\sigma_{\text{nonlin}}$ were the standard deviations of the linear and nonlinear terms alone.

For this problem we use random features of varying regular degrees $d = 1, 3, 10, 16$ and number of data points $n$. The features use a Fourier nonlinearity $h(\cdot) = (\sin \cdot, \cos \cdot)$, weights $w_{ij} \sim N(0, d^{-1/2})$, and biases $b_i \sim U([-\pi, \pi])$, leading to an RBF kernel in $d$ dimensions. The output regression model is again ridge regression with the penalty selected via cross-validation on the training set from 7 logarithmically spaced points between $10^{-4}$ and $10^2$.

In Figure 3, we show the test error as well as the selected ridge penalty for different values of $d$ and $n$. With a small amount of data ($n < 250$), the model with $d = 1$ has the lowest test error, since this "simplest" model is less likely to overfit. On the other hand, in the intermediate data regime ($250 < n < 400$), the model with $d = 3$ does best. For large amounts of data ($n > 400$), all of the models with interactions $d \geq 3$ do roughly the same. Note that with the RBF kernel the RKHS $\mathcal{H}_d \subseteq \mathcal{H}_{d'}$ whenever $d \leq d'$, so $d > 3$ can still capture the degree 3 polynomial model. However, we see that the more complex models have a higher ridge penalty selected. The penalty is able to adaptively control this complexity given enough data.

### A.2 Stability with respect to sparse input noise

Here we show that sparse random features are stable for spike-and-slab input noise. In this example, the truth follows a linear model, where we have random input points $\mathbf{x}_i \sim \mathcal{N}(0, \mathbf{I})$ and linear observations $y_i = \mathbf{x}_i^\mathsf{T} \beta$ for $i = 1, \ldots, n$ and $\beta \sim \mathcal{N}(0, \mathbf{I})$. However, we only have access to sparsely corrupted inputs $\mathbf{w}_i = \mathbf{x}_i + \mathbf{e}_i$, where $\mathbf{e}_i = 0$ with probability $1 - p$ and $\mathbf{e}_i = \epsilon_x - \mathbf{x}_i$ with probability $p$, $\epsilon_x \sim \mathcal{N}(0, \sigma^2 \mathbf{I})$. That is, the corrupted inputs are replaced with pure noise. We use $p = 0.03 \ll 1$ and $\sigma = 6 \gg 1$ so that the noise is sparse but large when it occurs.

In Table 1 we show the performance of various methods on this regression problem given the corrupted data $(\mathbf{W}, \mathbf{y})$. Note that if the practitioner has access to the uncorrupted data $\mathbf{X}$, linear regression succeeds with a perfect score of 1. Using kernel ridge regression with $k(\mathbf{x}, \mathbf{x}') = 1 - \frac{1}{l} \|\mathbf{x} - \mathbf{x}'\|_1$, the kernel that arises from sparse random features with $d = 1$ and sign nonlinearity, leads to improved performance over naïve linear regression on the corrupted data or a robust Huber loss function. The

| Model | Training score | Testing score |
|---|---|---|
| Linear | 0.854 | 0.453 |
| Kernel | 1.000 | 0.607 |
| Trim + linear | 0.945 | 0.686 |
| Huber | 0.858 | 0.392 |

Table 1: Scores ($R^2$ coefficient) of various regression models on linear data with corrupted inputs. In the presence of these errors, linear regression fails to acheive as good a test score as the kernel method, which is almost as good as trimming before performing regression and better than the robust Huber estimator.

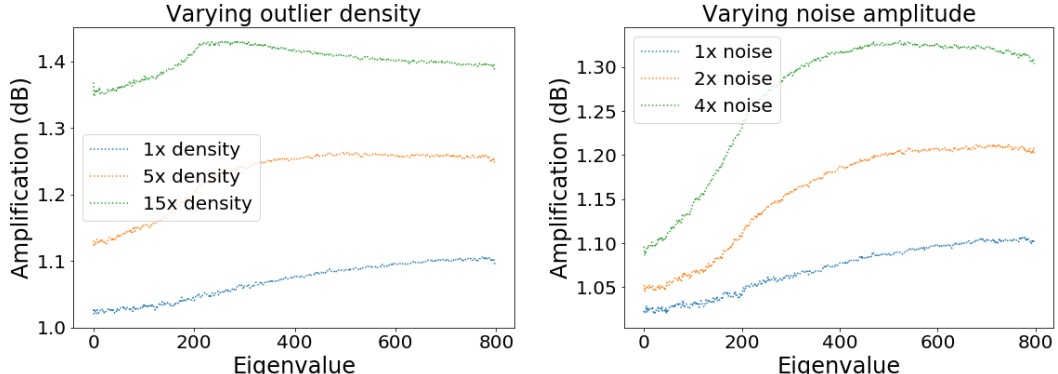

Figure 4: Kernel eigenvalue amplification while **(left)** varying $p$ with $\sigma = 6$ fixed, and **(right)** varying $\sigma$ with $p = 0.03$ fixed. Plotted is the ratio of eigenvalues of the kernel matrix corrupted by noise to those without any corruption, ordered from largest to smallest in magnitude. We see that the sparse feature kernel shows little noise amplification when it is sparse (right), even for large amplitude. On the other hand, less sparse noise does get amplified (left).

best performance is attained by trimming the outliers and then performing linear regression. However, this is meant to illustrate our point that sparse random features and their corresponding kernels may be useful when dealing with noisy inputs in a learning problem.

In Figure 4 we show another way of measuring this stability property. We compute the eigenvalues of the kernel matrix on a fixed dataset of size $n = 800$ points both with noise and without noise. Plotted are the ratio of the noisy to noiseless eigenvalues, in decibels, which we call the amplification and is a measure of how corrupted the kernel matrix is by this noise. The main trend that we see is, for fixed $p = 3$, changing the amplitude of the noise $\sigma$ does not lead to significant amplification, especially of the early eigenvalues which are of largest magnitude. On the other hand, making the outliers denser does lead to more amplification of all the eigenvalues. The eigenspace spanned by the largest eigenvalues is the most "important" for any learning problem.

## B    Kernel examples

### B.1    Fully-connected weights

We will now describe a number of common random features and the kernels they generate with fully-connected weights. Later on, we will see how these change as sparsity is introduced in the input-hidden connections.

**Translation invariant kernels**    The classical random features [2] sample Gaussian weights $\mathbf{w} \sim N(0, \sigma^{-2}\mathbf{I})$, uniform biases $b \sim U[-a, a]$, and employ the Fourier nonlinearity $h(\cdot) = \cos(\cdot)$. This leads to the Gaussian radial basis function kernel

$$k(\mathbf{x}, \mathbf{x}') = \exp\left(-\frac{1}{2\sigma^2}\|\mathbf{x} - \mathbf{x}'\|^2\right),$$

for $\mathbf{x}, \mathbf{x}' \in [-a, a]^l$. In fact, every translation-invariant kernel arises from Fourier nonlinearities for some distributions of weights and biases (Bôchner's theorem).

**Moment generating function kernels**  The exponential function is more similar to the kinds of monotone firing rate curves found in biological neurons. In this case, we have

$$k(\mathbf{x}, \mathbf{x}') = \mathbb{E} \exp(\mathbf{w}^\mathsf{T}(\mathbf{x} + \mathbf{x}') + 2b).$$

We can often evaluate this expectation using moment generating functions. For example, if $\mathbf{w}$ and $b$ are independent, which is a common assumption, then

$$k(\mathbf{x}, \mathbf{x}') = \mathbb{E} \left( \exp(\mathbf{w}^\mathsf{T}(\mathbf{x} + \mathbf{x}')) \cdot \mathbb{E} \exp(2b), \right.$$

where $\mathbb{E} \left( \exp(\mathbf{w}^\mathsf{T}(\mathbf{x} + \mathbf{x}')) \right)$ is the moment generating function for the marginal distribution of $\mathbf{w}$, and $\mathbb{E} \exp(2b)$ is just a constant that scales the kernel.

For multivariate Gaussian weights $\mathbf{w} \sim N(\mathbf{m}, \mathbf{\Sigma})$ this becomes

$$k(\mathbf{x}, \mathbf{x}') = \exp \left( \mathbf{m}^\mathsf{T}(\mathbf{x} + \mathbf{x}') + \frac{1}{2}(\mathbf{x} + \mathbf{x}')^\mathsf{T} \mathbf{\Sigma}(\mathbf{x} + \mathbf{x}') \right) \cdot \mathbb{E} \exp(2b).$$

This equation becomes more interpretable if $\mathbf{m} = 0$ and $\mathbf{\Sigma} = \sigma^{-2}\mathbf{I}$ and the input data are normalized: $\|\mathbf{x}\| = \|\mathbf{x}'\| = 1$. Then,

$$k(\mathbf{x}, \mathbf{x}') \propto \exp \left( \sigma^{-2} \mathbf{x}^\mathsf{T} \mathbf{x}' \right) \propto \exp \left( -\frac{1}{2\sigma^2} \|\mathbf{x} - \mathbf{x}'\|^2 \right).$$

This result highlights that dot product kernels $k(\mathbf{x}, \mathbf{x}') = v(\mathbf{x}^\mathsf{T} \mathbf{x}')$, where $v : \mathbb{R} \to \mathbb{R}$, are radial basis functions on the sphere $S^{l-1} = \{\mathbf{x} \in \mathbb{R}^l : \|\mathbf{x}\|_2 = 1\}$. The eigenbasis of these kernels are the spherical harmonics [3, 4].

**Arc-cosine kernels**  This class of kernels is also induced by monotone "neuronal" nonlinearities and leads to different radial basis functions on the sphere [3, 5, 6]. Consider standard normal weights $\mathbf{w} \sim N(0, \mathbf{I})$ and nonlinearities which are threshold polynomial functions

$$h(z) = \Theta(z) z^p$$

for $p \in \mathbb{Z}^+$, where $\Theta(\cdot)$ is the Heaviside step function. The kernel in this case is given by

$$k(\mathbf{x}, \mathbf{x}') = 2 \int_{\mathbb{R}^l} \Theta(\mathbf{w}^\mathsf{T} \mathbf{x}) \Theta(\mathbf{w}^\mathsf{T} \mathbf{x}') (\mathbf{w}^\mathsf{T} \mathbf{x})^p (\mathbf{w}^\mathsf{T} \mathbf{x}')^p \frac{e^{\frac{-\|\mathbf{w}\|^2}{2}}}{(2\pi)^{l/2}} \, \mathrm{d}\mathbf{w}$$

$$= \frac{1}{\pi} \|\mathbf{x}\|^p \|\mathbf{x}'\|^p J_p(\theta),$$

for a known function $J_p(\theta)$ where $\theta = \arccos \left( \frac{\mathbf{x}^\mathsf{T} \mathbf{x}'}{\|\mathbf{x}\| \|\mathbf{x}'\|} \right)$. Note that arc-cosine kernels are also dot product kernels. Also, if the weights are drawn as $\mathbf{w} \sim N(0, \sigma^{-2}\mathbf{I})$, the terms $\mathbf{x}$ are replaced by $\mathbf{x}/\sigma$, but this does not affect $\theta$. With $p = 0$, corresponding to the step function nonlinearity, we have $J_0(\theta) = \pi - \theta$, and the resulting kernel does not depend on $\|\mathbf{x}\|$ or $\|\mathbf{x}'\|$:

$$k(\mathbf{x}, \mathbf{x}) = 1 - \frac{1}{\pi} \arccos \left( \frac{\mathbf{x}^\mathsf{T} \mathbf{x}'}{\|\mathbf{x}\| \|\mathbf{x}'\|} \right). \tag{7}$$

**Sign nonlinearity**  We also consider a shifted version of the step function nonlinearity, the sign function $\mathrm{sgn}(z)$, equal to $+1$ when $z > 0$, $-1$ when $z < 0$, and zero when $z = 0$. Let $b \sim U([a_1, a_2])$ and $\mathbf{w} \sim P$, where $P$ is any spherically symmetric distribution, such as a Gaussian. Then,

$$k(\mathbf{x}, \mathbf{x}') = \mathbb{E} \left[ \int_{a_1}^{a_2} \frac{\mathrm{d}b}{a_2 - a_1} \, \mathrm{sgn}(\mathbf{w}^\mathsf{T} \mathbf{x} - b) \, \mathrm{sgn}(\mathbf{w}^\mathsf{T} \mathbf{x}' - b) \right]$$

$$= \mathbb{E} \left[ 1 - 2 \frac{|\mathbf{w}^\mathsf{T} \mathbf{x} - \mathbf{w}^\mathsf{T} \mathbf{x}'|}{a_2 - a_1} \right]$$

$$= 1 - \frac{2}{a_2 - a_1} \mathbb{E} |\mathbf{w}^\mathsf{T}(\mathbf{x} - \mathbf{x}')|$$

$$= 1 - 2\mathbb{E}(|\mathbf{w}^\mathsf{T} \mathbf{e}|) \frac{\|\mathbf{x} - \mathbf{x}'\|_2}{a_2 - a_1}$$

where $\mathbf{e} = (\mathbf{x} - \mathbf{x}')/\|\mathbf{x} - \mathbf{x}'\|_2$. The factor $\mathbb{E}(|\mathbf{w}^\mathsf{T}\mathbf{e}|)$ in front of the norm is just a function of the radial part of the distribution $P$, which we should set inversely proportional to $\sqrt{l}$ to match the scaling of $\|\mathbf{x} - \mathbf{x}'\|_2$. For $\mathbf{w} \sim N(0, \sigma^2 l^{-1}\mathbf{I})$, we obtain

$$k(\mathbf{x}, \mathbf{x}') = 1 - 2\sigma\sqrt{\frac{2}{\pi l}} \frac{\|\mathbf{x} - \mathbf{x}'\|_2}{a_2 - a_1}. \tag{8}$$

### B.2 Sparse weights

The sparsest networks possible have $d = 1$, leading to first-order additive kernels. Here we look at two simple nonlinearities where we can perform the sum and obtain an explicit formula for the additive kernel. In both cases, the kernels are simply related to a robust distance metric. This suggests that such kernels may be useful in cases where there are outlier coordinates in the input data.

**Step function nonlinearity**  We again consider the step function nonlinearity $h(\cdot) = \Theta(\cdot)$, which in the case of fully-connected Gaussian weights leads to the degree $p = 0$ arc-cosine kernel $k(\mathbf{x}, \mathbf{x}') = 1 - \frac{\theta(\mathbf{x}, \mathbf{x}')}{\pi}$. When $d = 1$, $\mathbf{x}_\mathcal{N} = x_i$ and $\mathbf{x}'_\mathcal{N} = x'_i$ are scalars. For a scalar $a$, normalization leads to $a/\|a\| = \text{sgn}(a)$. Therefore, $\theta = \arccos(\text{sgn}(x_i)\text{sgn}(x'_i)) = 0$ if $\text{sgn}(x_i) = \text{sgn}(x'_i)$ and $\pi$ otherwise. Performing the sum in (3), we find that the kernel becomes

$$k_1^{\text{reg}}(\mathbf{x}, \mathbf{x}') = 1 - \frac{|\{i : \text{sgn}(x_i) \neq \text{sgn}(x'_i)\}|}{l} = 1 - \frac{\|\text{sgn}(\mathbf{x}) - \text{sgn}(\mathbf{x}')\|_0}{l}. \tag{9}$$

This kernel is equal to one minus the normalized Hamming distance of vectors $\text{sgn}(\mathbf{x})$ and $\text{sgn}(\mathbf{x}')$. The fully-connected kernel, on the other hand, uses the full angle between the vectors $x$ and $x'$. The sparsity can be seen as inducing a "quantization," via the sign function, on these vectors. Finally, if the data are in the binary hypercube, with $\mathbf{x}$ and $\mathbf{x}' \in \{-1, +1\}^l$, then the kernel is exactly one minus the normalized Hamming distance.

**Sign nonlinearity**  We now consider a slightly different nonlinearity, the sign function. It will turn out that the kernel is quite different than for the step function. This has $h(\cdot) = \text{sgn}(\cdot) = 2\Theta(\cdot) - 1$. Let $b \sim U([a_1, a_2])$ and $w \sim P$. Then,

$$\begin{aligned} k_1^{\text{reg}}(\mathbf{x}, \mathbf{x}') &= \frac{1}{l}\sum_{i=1}^{l} \mathbb{E}_P\left[\int_{a_1}^{a_2} \frac{\mathrm{d}b}{a_2 - a_1} \text{sgn}(wx_i - b)\text{sgn}(wx'_i - b)\right] \\ &= \frac{1}{l}\sum_{i=1}^{l} \mathbb{E}_P\left[1 - 2\frac{|wx_i - wx'_i|}{a_2 - a_1}\right] \\ &= 1 - \frac{2\mathbb{E}_P(|w|)}{l}\frac{\|x - x'\|_1}{a_2 - a_1}. \end{aligned} \tag{10}$$

Choosing $P(w) = \frac{1}{2}\delta(w + 1) + \frac{1}{2}\delta(w - 1)$ and $a_2 = -a_1 = a$ recovers the "random stump" result of Rahimi and Recht [2]. Despite the fact that sign is just a shifted version of the step function, the kernels are quite different: the sign nonlinearity does not exhibit the quantization effect and depends on the $\ell^1$-norm rather than the $\ell^0$-"norm".

## C  Kernel approximation results

We now show a basic uniform convergence result for any random features, not necessarily sparse, that use Lipschitz continuous nonlinearities. Recall the definition of a Lipschitz function:

**Definition 1.** *A function $f : \mathcal{X} \to \mathbb{R}$ is said to be $L$-**Lipschitz continuous** (or Lipschitz with constant $L$) if*

$$|f(\mathbf{x}) - f(\mathbf{y})| \leq L\|\mathbf{x} - \mathbf{y}\|$$

*holds for all $\mathbf{x}, \mathbf{y} \in \mathcal{X}$. Here, $\|\cdot\|$ is a norm on $\mathcal{X}$ (the $\ell^2$-norm unless otherwise specified).*

Assuming that $h$ is Lipschitz and some regularity assumptions on the distribution $\mu$, the random feature expansion approximates the kernel uniformly over $\mathcal{X}$. As far as we are aware, this result has

not been stated previously, although it appears to be known (see Bach [7]) and is very similar to Claim 1 in Rahimi and Recht [2] which holds only for random Fourier features (see also Sutherland and Schneider [8] and Sriperumbudur and Szabo [9] for improved results in this case). The rates we obtain for Lipschitz nonlinearities are not essentially different than those obtained in the Fourier features case.

As for the examples we have given, the only ones which are not Lipschitz are the step function (order 0 arc-cosine kernel) and sign nonlinearities. Since these functions are discontinuous, their convergence to the kernel occurs in a weaker than uniform sense. However, our result does apply to the rectified linear nonlinearity (order 1 arc-cosine kernel), which is non-differentiable at zero but 1-Lipschitz and widely applied in artificial neural networks. The proof of the following Theorem appears at the end of this section.

**Theorem 1** (Kernel approximation for Lipschitz nonlinearities). *Assume that $\mathbf{x} \in \mathcal{X} \subset \mathbb{R}^l$ and that $\mathcal{X}$ is compact, $\Delta = \operatorname{diam}(\mathcal{X})$, and the null vector $0 \in \mathcal{X}$. Let the weights and biases $(\mathbf{w}, b)$ follow the distribution $\mu$ on $\mathbb{R}^{l+1}$ with finite second moments. Let $h(\cdot)$ be a nonlinearity which is $L$-Lipschitz continuous and define the random feature $\phi : \mathbb{R}^l \to \mathbb{R}$ by $\phi(\mathbf{x}) = h(\mathbf{w}^\intercal \mathbf{x} - b)$. We assume that the following hold for all $\mathbf{x} \in \mathcal{X}$: $|\phi(\mathbf{x})| \leq \kappa$ almost surely, $\mathbb{E}\,|\phi(\mathbf{x})|^2 < \infty$, and $\mathbb{E}\,\phi(\mathbf{x})\phi(\mathbf{x}') = k(\mathbf{x}, \mathbf{x}')$.*

*Then $\sup_{\mathbf{x},\mathbf{x}' \in \mathcal{X}} \left| \frac{1}{m}\phi(\mathbf{x})^\intercal \phi(\mathbf{x}') - k(\mathbf{x}, \mathbf{x}') \right| \leq \epsilon$ with probability at least*

$$1 - 2^8 \left( \frac{\kappa L \Delta \sqrt{\mathbb{E}\|\mathbf{w}\|^2 + 3(\mathbb{E}\|\mathbf{w}\|)^2}}{\epsilon} \right)^2 \exp\left( \frac{-m\epsilon^2}{8(l+1)\kappa^2} \right).$$

**Sample complexity**  Theorem 1 guarantees uniform approximation up to error $\epsilon$ using $m = O\left( \frac{l\kappa^2}{\epsilon^2} \log \frac{C}{\epsilon} \right)$ features. This is precisely the same dependence on $l$ and $\epsilon$ as for random Fourier features. Note that [10] also found that $m$ should scale linearly with $l$ to minimize error in a particular classification task.

A limitation of Theorem 1 is that it only shows approximation of the limiting kernel rather than direct approximation of functions in the RKHS. A more detailed analysis of the convergence to RKHS is contained in the work of Bach [7], whereas Rudi and Rosasco [11] directly analyze the generalization ability of these approximations. Sun et al. [12] show even faster rates which also apply to SVMs, assuming that the features are compatible ("optimized") for the learning problem. Also, the techniques of Sutherland and Schneider [8] and Sriperumbudur and Szabo [9] could be used to improve our constants and prove convergence in other $L^p$ norms.

In the sparse case, we must extend our probability space to capture the randomness of (1) the degrees, (2) the neighborhoods conditional on the degree, and (3) the weight vectors conditional on the degree and neighborhood. The degrees are distributed independently according to $d_i \sim D$, with some abuse of notation since we also use $D(d)$ to represent the probability mass function. We shall always think of the neighborhoods $\mathcal{N} \sim \nu|d$ as chosen uniformly among all $d$ element subsets, where $\nu|d$ represents this conditional distribution. Finally, given a neighborhood of some degree, the nonzero weights and bias are drawn from a distribution $(\mathbf{w}, b) \sim \mu|d$ on $\mathbb{R}^{d+1}$. For simpler notation, we do not show any dependence on the neighborhood here, since we will always take the actual weight values to not depend on the particular neighborhood $\mathcal{N}$. However, strictly speaking, the weights do depend on $\mathcal{N}$ because that determines their support. Finally, we use $\mathbb{E}$ to denote expectation over all variables (degree, neighborhood, and weights), whereas we use $\mathbb{E}_{\mu|d}$ for the expectation under $\mu|d$ for a given degree.

**Corollary 2** (Kernel approximation with sparse features). *Assume that $\mathbf{x} \in \mathcal{X} \subset \mathbb{R}^l$ and that $\mathcal{X}$ is compact, $\Delta = \operatorname{diam}(\mathcal{X})$, and the null vector $0 \in \mathcal{X}$. Let the degrees $d$ follow the degree distribution $D$ on $[l]$. For every $d \in [l]$, let $\mu|d$ denote the conditional distributions for $(\mathbf{w}, b)$ on $\mathbb{R}^{d+1}$ and assume that these have finite second moments. Let $h(\cdot)$ be a nonlinearity which is $L$-Lipschitz continuous, and define the random feature $\phi : \mathbb{R}^l \to \mathbb{R}$ by $\phi(\mathbf{x}) = h(\mathbf{w}^\intercal \mathbf{x} - b)$, where $\mathbf{w}$ follows the degree distribution model. We assume that the following hold for all $\mathbf{x}_\mathcal{N} \in \mathcal{X}_\mathcal{N}$ with $|\mathcal{N}| = d$, and for all $1 \leq d \leq l$: $|\phi(\mathbf{x}_\mathcal{N})|^2 \leq \kappa$ almost surely under $\mu|d$, $\mathbb{E}\left[|\phi(\mathbf{x}_\mathcal{N})|^2|d\right] < \infty$, and $\mathbb{E}[\phi(\mathbf{x}_\mathcal{N})\phi(\mathbf{x}'_\mathcal{N})|d] = k_d^{\operatorname{reg}}(\mathbf{x}_\mathcal{N}, \mathbf{x}'_\mathcal{N})$.*

*Then* $\sup_{\mathbf{x},\mathbf{x}'\in\mathcal{X}} \left| \frac{1}{m}\boldsymbol{\phi}(\mathbf{x})^\mathsf{T}\boldsymbol{\phi}(\mathbf{x}') - k_D^{\mathrm{dist}}(\mathbf{x},\mathbf{x}') \right| \le \epsilon$, *with probability at least*

$$1 - 2^8 \left( \frac{\kappa L \Delta \sqrt{\mathbb{E}\|\mathbf{w}\|^2 + 3(\mathbb{E}\|\mathbf{w}\|)^2}}{\epsilon} \right)^2 \exp\left( \frac{-m\epsilon^2}{8(l+1)\kappa^2} \right).$$

*The kernels $k_d^{\mathrm{reg}}(\mathbf{z},\mathbf{z}')$ and $k_D^{\mathrm{dist}}(\mathbf{x},\mathbf{x}')$ are given by equations* (3) *and* (4).

*Proof.* It suffices to show that conditions (1–3) on the conditional distributions $\mu|d$, $d \in [l]$, imply conditions (1–3) in Theorem 1. Conditions (1) and (2) clearly hold, since the distribution $D$ has finite support. By construction, $\mathbb{E}\,\phi(\mathbf{x})\phi(\mathbf{x}') = \mathbb{E}[\mathbb{E}[\phi(\mathbf{x}_\mathcal{N})\phi(\mathbf{x}'_\mathcal{N})|d]] = \mathbb{E}[k_d^{\mathrm{reg}}(\mathbf{x}_\mathcal{N},\mathbf{x}'_\mathcal{N})] = k_D^{\mathrm{dist}}(\mathbf{x},\mathbf{x}')$, which concludes the proof. $\qquad\square$

**Differences of sparsity** The only difference we find with sparse random features is in the terms $\mathbb{E}\|\mathbf{w}\|^2$ and $\mathbb{E}\|\mathbf{w}\|$, since sparsity adds variance to the weights. This suggests that scaling the weights so that $\mathbb{E}_{\mu|d}\|\mathbf{w}\|^2$ is constant for all $d$ is a good idea. For example, setting $(\mathbf{w}_i)_{\mathcal{N}_i} \sim N(0, \sigma^2 d_i^{-1}\mathbf{I}_{d_i})$, the random variables $\|\mathbf{w}_i\|^2 \sim \sigma^2 d_i^{-1}\chi^2(d_i)$ and $\|\mathbf{w}_i\| \sim \sigma d_i^{-1/2}\chi(d_i)$. Then $\mathbb{E}\|\mathbf{w}_i\|^2 = \sigma^2$ irregardless of $d_i$ and $\mathbb{E}\|\mathbf{w}_i\| = \sigma(1 + o(d_i))$. With this choice, the number of sparse features needed to achieve an error $\epsilon$ is the same as in the dense case, up to a small constant factor. This is perhaps remarkable since there could be as many as $2^l$ terms in the expression of $k_D^{\mathrm{dist}}(\mathbf{x},\mathbf{x}')$. However, the random feature expansion does not need to approximate all of these terms well, just their average.

*Proof of Theorem 1.* We follow the approach of Claim 1 in [2], a similar result for random Fourier features but which crucially uses the fact that the trigonometric functions are differentiable and bounded. For simplicity of notation, let $\boldsymbol{\xi} = (\mathbf{x}, \mathbf{x}')$ and define the *direct sum norm* on $\mathcal{X}^+ = \mathcal{X} \oplus \mathcal{X}$ as $\|\boldsymbol{\xi}\|_+ = \|\mathbf{x}\| + \|\mathbf{x}'\|$. Under this norm $\mathcal{X}^+$ is a Banach space but not a Hilbert space, however this will not matter. For $i = 1, \ldots, m$, let

$$f_i(\boldsymbol{\xi}) = \phi_i(\mathbf{x})\phi_i(\mathbf{x}'),$$
$$g_i(\boldsymbol{\xi}) = \phi_i(\mathbf{x})\phi_i(\mathbf{x}') - k(\mathbf{x},\mathbf{x}')$$
$$= f_i(\boldsymbol{\xi}) - \mathbb{E}f_i(\boldsymbol{\xi}),$$

and note that these $g_i$ are i.i.d., centered random variables. By assumptions (1) and (2), $f_i$ and $g_i$ are absolutely integrable and $k(\mathbf{x},\mathbf{x}') = \mathbb{E}\,\phi_i(\mathbf{x})\phi_i(\mathbf{x}')$. Denote their mean by

$$\bar{g}(\boldsymbol{\xi}) = \frac{1}{m}\boldsymbol{\phi}(\mathbf{x})^\mathsf{T}\boldsymbol{\phi}(\mathbf{x}') - k(\mathbf{x},\mathbf{x}') = \frac{1}{m}\sum_{i=1}^m g_i(\boldsymbol{\xi}).$$

Our goal is to show that $|\bar{g}(\boldsymbol{\xi})| \le \epsilon$ for all $\boldsymbol{\xi} \in \mathcal{X}^+$ with sufficiently high probability.

The space $\mathcal{X}^+$ is compact and $2l$-dimensional, and it has diameter at most twice the diameter of $\mathcal{X}$ under the sum norm. Thus we can cover $\mathcal{X}^+$ with an $\epsilon$-net using at most $T = (4\Delta/R)^{2l}$ balls of radius $R$ [13]. Call the centers of these balls $\boldsymbol{\xi}_i$ for $i = 1, \ldots, T$, and let $\bar{L}$ denote the Lipschitz constant of $\bar{g}$ with respect to the sum norm. Then we can show that $|\bar{g}(\boldsymbol{\xi})| \le \epsilon$ for all $\boldsymbol{\xi} \in \mathcal{X}^+$ if we show that

1. $\bar{L} \le \frac{\epsilon}{2R}$, and

2. $|\bar{g}(\boldsymbol{\xi}_i)| \le \frac{\epsilon}{2}$ for all $i$.

First, we bound the Lipschitz constant of $g_i$ with respect to the sum norm $\|\cdot\|_+$. Since $h$ is $L$-Lipschitz, we have that $\phi_i$ is Lipschitz with constant $L\|\mathbf{w}_i\|$. Thus, letting $\boldsymbol{\xi}' = \boldsymbol{\xi} + (\boldsymbol{\delta}, \boldsymbol{\delta}')$,

$$2|f_i(\boldsymbol{\xi}) - f_i(\boldsymbol{\xi}')| \le |\phi_i(\mathbf{x}+\boldsymbol{\delta})\phi_i(\mathbf{x}'+\boldsymbol{\delta}') - \phi_i(\mathbf{x}+\boldsymbol{\delta})\phi_i(\mathbf{x}')|$$
$$+ |\phi_i(\mathbf{x}+\boldsymbol{\delta})\phi_i(\mathbf{x}'+\boldsymbol{\delta}') - \phi_i(\mathbf{x})\phi_i(\mathbf{x}'+\boldsymbol{\delta}')|$$
$$+ |\phi_i(\mathbf{x}+\boldsymbol{\delta})\phi_i(\mathbf{x}') - \phi_i(\mathbf{x})\phi_i(\mathbf{x}')|$$
$$+ |\phi_i(\mathbf{x})\phi_i(\mathbf{x}'+\boldsymbol{\delta}') - \phi_i(\mathbf{x})\phi_i(\mathbf{x}')|$$
$$\le 2L\|\mathbf{w}_i\| \cdot \sup_{\mathbf{x}\in\mathcal{X}} |\phi_i(\mathbf{x})| \cdot (\|\boldsymbol{\delta}\| + \|\boldsymbol{\delta}'\|)$$
$$= 2\kappa L\|\mathbf{w}_i\| \cdot \|\boldsymbol{\xi} - \boldsymbol{\xi}'\|_+,$$

we have that $f_i$ has Lipschitz constant $\kappa L\|\mathbf{w}_i\|$. This implies that $g_i$ has Lipschitz constant $\leq \kappa L(\|\mathbf{w}_i\| + \mathbb{E}\|\mathbf{w}\|)$.

Let $\bar{L}$ denote the Lipschitz constant of $\bar{g}$. Note that $\mathbb{E}\bar{L} \leq 2\kappa L\mathbb{E}\|\mathbf{w}\|$. Also,

$$\mathbb{E}\bar{L}^2 \leq L^2\kappa^2\mathbb{E}\left(\|\mathbf{w}\| + \mathbb{E}\|\mathbf{w}\|\right)^2$$
$$= L^2\kappa^2\left(\mathbb{E}\|\mathbf{w}\|^2 + 3(\mathbb{E}\|\mathbf{w}\|)^2\right).$$

Markov's inequality states that $\Pr[\bar{L}^2 > t^2] \leq \mathbb{E}[\bar{L}^2]/t^2$. Letting $t = \frac{\epsilon}{2R}$, we find that

$$\Pr[\bar{L} > t] = \Pr\left[\bar{L} > \frac{\epsilon}{2R}\right] \leq L^2\kappa^2\left(\mathbb{E}\|\mathbf{w}\|^2 + 3(\mathbb{E}\|\mathbf{w}\|)^2\right)\left(\frac{2R}{\epsilon}\right)^2. \tag{11}$$

Now we would like to show that $|\bar{g}(\boldsymbol{\xi}_i)| \leq \epsilon/2$ for all $i = 1, \ldots, T$ anchors in the $\epsilon$-net. A straightforward application of Hoeffding's inequality and a union bound shows that

$$\Pr\left[|\bar{g}(\boldsymbol{\xi}_i)| > \frac{\epsilon}{2} \text{ for all } i\right] \leq 2T\exp\left(\frac{-m\epsilon^2}{8\kappa^4}\right), \tag{12}$$

since $|f_i(\boldsymbol{\xi})| \leq \kappa^2$.

Combining equations (11) and (12) results in a probability of failure

$$\Pr\left[\sup_{\boldsymbol{\xi}\in\mathcal{X}^+}|\bar{g}(\boldsymbol{\xi})| \geq \epsilon\right] \leq 2\left(\frac{4\Delta}{R}\right)^{2l}\exp\left(\frac{-m\epsilon^2}{8\kappa^2}\right) + L^2\kappa^2(\mathbb{E}\|\mathbf{w}\|^2 + 3(\mathbb{E}\|\mathbf{w}\|)^2)\left(\frac{2R}{\epsilon}\right)^2$$
$$= aR^{-2l} + bR^2. \tag{13}$$

Set $R = (a/b)^{\frac{1}{2l+2}}$, so that the probability (13) has the form, $2a^{\frac{2}{2l+2}}b^{\frac{2l}{2l+2}}$. Thus the probability of failure satisfies

$$\Pr\left[\sup_{\boldsymbol{\xi}\in\mathcal{X}^+}|\bar{g}(\boldsymbol{\xi})| \geq \epsilon\right] \leq 2a^{\frac{2}{2l+2}}b^{\frac{2l}{2l+2}}$$

$$= 2 \cdot 2^{\frac{2}{2l+2}}\left(\frac{8\kappa L\Delta\sqrt{\mathbb{E}\|\mathbf{w}\|^2 + 3(\mathbb{E}\|\mathbf{w}\|)^2}}{\epsilon}\right)^{\frac{4l}{2l+2}}\exp\left(\frac{-m\epsilon^2}{4(2l+2)\kappa^2}\right)$$

$$\leq 2^8\left(\frac{\kappa L\Delta\sqrt{\mathbb{E}\|\mathbf{w}\|^2 + 3(\mathbb{E}\|\mathbf{w}\|)^2}}{\epsilon}\right)^2\exp\left(\frac{-m\epsilon^2}{8(l+1)\kappa^2}\right),$$

for all $l \in \mathbb{N}$, assuming $\Delta\kappa L\sqrt{\mathbb{E}\|\mathbf{w}\|^2 + 3(\mathbb{E}\|\mathbf{w}\|)^2} > \epsilon$. Considering the complementary event concludes the proof. $\square$

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
