# OpenReview forum: "Additive function approximation in the brain"
_NeurIPS.cc/2019/Workshop/Neuro_AI — Real Neurons & Hidden Units @ NeurIPS 2019 Poster_

### Official Review · AnonReviewer3 · 2019-09-22
**An interesting and possibly important approach, but its novelty and significance are not clear**

**Clarity:** 2

**Comment:**

The overall subject of this paper seems important and novel enough to merit discussion at this workshop, but as it currently stands this paper does not clearly establish its contribution to this workshop. That said, I believe it is a simple matter of addressing the critiques made above. It is understandable that much of this could not be addressed within the four-page limit. In particular, it is my view that the connection to the kernel methods literature needs to be stated more clearly so as to be accessible to the more neuroscience-oriented audience of this workshop. The novel contribution of additive kernels to the theory of sparse random networks - a topic studied at length in the theoretical neuroscience literature - needs to be made explicit and clear.

**Category:**

AI->Neuro

**Clarity Comment:**

It seems like there are two separate ideas in this paper:
1) sparse random features are useful in feed-forward networks (section 4)
2) sparse random features implement additive kernels
The novelty of this paper presumably lies in the connection between these two ideas. While each of these was clearly explained, their connection was not at all clear to this reviewer. It is possible this is due to limited knowledge of the kernel methods literature.

**Evaluation:**

3: Good

**Importance:**

3: Important

**Importance Comment:**

Many of the arguments made for why sparse random features are useful in the brain (namely, section 4) have been previously discussed in the literature (namely, refs 20,44,45). The novel contribution of this paper seems to be the connection to additive kernel methods, but it was not clear what this added to the previous theory on sparse and random networks. It is critical that this be made explicit for this work to comprise a valuable contribution to this workshop.

**Intersection:**

5: Outstanding

**Intersection Comment:**

This paper tries to link the theory of sparse random features and additive kernel functions to feed-foward networks in the brain. The introduction section elegantly reviews previous research on these topics, attempting to directly link experimental findings in the brain to the sparse random feature architecture motivated by machine learning theory and adopted in their proposed model.

**Rigor Comment:**

Within the four-page limit, no explicit demonstrations of their arguments from section 4 were made. However, extensive connections to previous literature in the machine learning and theoretical neuroscience literature are made, making their arguments relatively convincing. A critical piece missing is explicit connections to previous theory on this subject. For example, how do their scaling results on the number of neurons (namely, the "claim" in section 3) relate to those of reference 45?

**Technical Rigor:**

3: Convincing

---

### Official Review · AnonReviewer1 · 2019-09-26
**Interesting connections between sparse connectivity and kernel literature**

**Clarity:** 4

**Comment:**

This submission investigates potential advantages of sparsity, first showing that kernels induced by sparse features can be approximated with a number of random features that grows linearly in the input dimension, then connects sparse kernels to literature on additive modeling and suggests stability and scalability benefits. The implication is that the sparse nature of connectivity in the brain has advantages beyond merely satisfying biological constraints. Although not totally conclusive, I believe that this work has the potential to provoke interesting discussion.

Strengths:

The work seems to be mathematically rigorous and the conclusions drawn are interesting, if not totally conclusive.

The submission is generally quite readable given the space restrictions.

Weaknesses:

Intuitively, I'm not sure the kernels discussed here will be universal kernels.

I'm not totally convinced by the robustness arguments in Section 4.2.

Although not strictly necessary, especially for a workshop submission, no experimental demonstration of the suggested advantages of sparse connectivity is provided.

The Call for Papers for this conference states that submissions should not be more than 4 pages long including references and appendices. Although the main text of this submission is 4 pages long, it has 2 pages of references and a 9 page appendix. I did not read the appendix.

**Category:**

Neuro->AI

**Clarity Comment:**

I think this paper is about as readable as it can be given the space restriction. I am not extremely familiar with literature on random features approximations or generalized additive models, but I still found the arguments here to be relatively easy to read and grasp intuitively.

**Evaluation:**

4: Very good

**Importance:**

3: Important

**Importance Comment:**

Many machine learning methods implicitly assume models where all features are allowed to interact with each other. As far as I can tell, the main contribution of this work is an approximation guarantee for kernels induced by sparse connectivity. The submission also discusses further advantages of such kernels in terms of previous literature. Although none of these contributions seem groundbreaking, I believe that the connections drawn are interesting and useful.

**Intersection:**

5: Outstanding

**Intersection Comment:**

This article uses methods and results from the kernel approximation and generalized additive modeling literature to show benefits to sparsity. It is generally difficult to dissociate aspects of biological brains that serve important roles in terms of inductive bias or efficiency from aspects that arise from biological constraints. This kind of work plays a critical role here.

**Rigor Comment:**

I believe that this work is technically rigorous. I have not attempted to check the claim given in Section 3 as the proofs are well beyond the 4 page limit. The results in Section 4 seem to draw heavily on previously published work.

I'm not totally convinced by the arguments in Section 4.2. If the regressor relies heavily on a handful of features and e is aligned with these features, e might still substantially change f, so I don't see any inherent reason that sparsity should lead to adversarial robustness.  Moreover, the treatment here seems to cover sparse attacks (e.g. adversarial examples constructed with respect to L_1 or L_0 norms) but most literature on adversarial attacks treats adversarial examples constructed vs. L_\inf and L_2 norms, where it is unclear whether there is any advantage of sparsity.

**Technical Rigor:**

3: Convincing

---

### Official Review · AnonReviewer2 · 2019-09-26
**Review of sparse connectivity and kernel functions**

**Clarity:** 4

**Comment:**

The biological links presented in this work are interesting. The authors also do well in acknowledging the limitations of this model for biological application (for example that cell types and neuromodulator levels are an important feature of biological networks).

The appendix was quite frankly, long - beyond what I assume the limitations of the conference organizers intended. Although this information was likely helpful in explaining the details of the method, I did not fully cover the entirety of the appendix.

**Category:**

AI->Neuro

**Clarity Comment:**

The ideas contained here are contextualized to their biological counterparts very well by the authors.

**Evaluation:**

4: Very good

**Importance:**

3: Important

**Importance Comment:**

I am not completely sure how much value is added by the work presented here, relative to previous kernel work. However, this is still potentially interesting if framed within the context of prior literature.

**Intersection:**

4: High

**Intersection Comment:**

Links between AI (sparse connectivity) and the nature of biological connectivity are drawn throughout. This was well done. I appreciate the specific examples detailed by the authors (mushroom bodies). The limitations of the approach with respect to the context of biological networks, i.e. neuromodulatory cells, etc. were presented, which is a helpful point to raise when comparing any artificial system to a biological one.

**Rigor Comment:**

The work appears technically rigorous, albeit highly based on the previous literature.

**Technical Rigor:**

4: Very convincing

---

### Decision · Program_Chairs · 2019-10-02

Accept (Poster)